# Estimating disorder probability based on polygenic prediction using the BPC approach

Emil Uffelmann [1] ✉, Major Depressive Disorder Working Group of the Psychiatric Genomics Consortium*, Schizophrenia Working Group of the Psychiatric Genomics Consortium*, Alkes L. Price [2,3,4], Danielle Posthuma[1,5] & Wouter J. Peyrot[1,6,7] ✉

Polygenic Scores (PGSs) summarize an individual's genetic propensity for a given trait. Bayesian methods, which improve the prediction accuracy of PGSs, are not well-calibrated for binary disorder traits in ascertained samples. This is a problem because well-calibrated PGSs are needed for future clinical implementation. We introduce the Bayesian polygenic score Probability Conversion (BPC) approach, which computes an individual's predicted disorder probability using genome-wide association study summary statistics, an existing Bayesian PGS method (e.g. PRScs, SBayesR), the individual's genotype data, and a prior disorder probability (which can be specified flexibly, based for example on literature, small reference samples, or prior elicitation). The BPC approach is practical in its application as it does not require a tuning sample with both genotype and phenotype data. Here, we show in simulated and empirical data of nine disorder traits that BPC yields well-calibrated results that are consistently better than the results of another recently published approach.

Polygenic Scores (PGSs)[1] are per-individual estimates of the total contribution of common genetic variants to a trait or disorder liability based on SNP effect sizes (betas) from Genome-Wide Association Studies (GWAS)[2]. PGSs for several traits show increasing clinical potential that rivals that of conventional clinical predictors[3–6]. While summarizing an individual's genetic risk for a disorder in a single value has the potential to be a simple and informative metric, PGS applications are limited because they are generally only interpretable at the group level. Accordingly, PGSs are commonly evaluated using the coefficient of determination ($R^2$)[7] or the Area Under the Curve (AUC)[8], metrics that are blind to the scale of the PGS. Moreover, risk estimates based on PGSs are often reported in quantiles (e.g., a PGS falls in the

top 5% of a given distribution), which can be challenging to interpret in terms of personal absolute risk of disease.

To make PGSs directly interpretable to individuals, they can be transformed into probabilities. For example, if an individual receives a PGS of 0.5 for multiple sclerosis, then this should correspond to a 50% probability of that individual developing multiple sclerosis in their lifetime. With access to a sufficiently large population-representative tuning sample with relevant pheno- and genotype data, such a transformation can be achieved with existing methods[9,10]. However, in most clinical settings, such samples are not readily available. Ideally, a single individual's genotype data and publicly available resources should be sufficient to achieve such a transformation.

[1]Department of Complex Trait Genetics, Center for Neurogenomics and Cognitive Research, Amsterdam Neuroscience, Vrije Universiteit Amsterdam, Amsterdam, The Netherlands. [2]Department of Epidemiology, Harvard T. H. Chan School of Public Health, Boston, MA, USA. [3]Department of Biostatistics, Harvard T. H. Chan School of Public Health, Boston, MA, USA. [4]Program in Medical and Population Genetics, Broad Institute of MIT and Harvard, Cambridge, MA, USA. [5]Department of Child and Adolescent Psychiatry and Pediatric Psychology, Section Complex Trait Genetics, Amsterdam Neuroscience, Vrije Universiteit Medical Center, Amsterdam University Medical Center, Amsterdam, The Netherlands. [6]Department of Psychiatry, Amsterdam, UMC, The Netherlands. [7]Amsterdam Public Health, Amsterdam, UMC, The Netherlands. *Lists of authors and their affiliations appear at the end of the paper. A full list of members and their affiliations appears in the Supplementary Information. ✉e-mail: e.uffelmann@vu.nl; w.peyrot@amsterdamumc.nl

Bayesian PGS methods are known to be well-calibrated for continuous traits[11–13], meaning the slope equals 1 when regressing the true phenotype on the PGS (implying the predicted values are, on average, equal to the true trait values). This offers a unique opportunity to achieve well-calibrated probabilities for binary disorder traits. However, when samples are over-ascertained for cases, Bayesian PGSs can become miscalibrated and, therefore, require a transformation.

Here, we introduce Bayesian polygenic score Probability Conversion (BPC), an approach to transform PGSs based on Bayesian methods (e.g. PRScs[12] and SBayesR[11]), that only requires a single individual's genotype data, GWAS summary statistics, and a prior disorder probability. We confirm that the resulting probabilities are well-calibrated in simulations and empirical analyses of nine disorders and that the BPC approach performs better than a recently published approach[14].

## Results

### Overview of methods

The BPC approach estimates absolute disorder probabilities using PGSs derived from GWAS summary statistics, genotype data, and a prior disorder probability, while avoiding the need for phenotype-informed tuning samples. BPC is designed to yield well-calibrated probabilities even in ascertained samples (i.e., when the risk is larger than the population prevalence), enabling its potential use in clinical contexts. The code to apply the BPC approach is publicly available (https://doi.org/10.5281/zenodo.15721084).

The BPC approach consists of 4 steps (see Fig. 1 and Methods). As input, BPC requires an individual's genotype data, a prior disorder probability (which can be informed by literature, small reference samples, or prior elicitation[15]), and GWAS summary statistics and their effective sample size ($N_{eff}$, see Supplementary Note 1). Additionally, it requires the population lifetime prevalence of the disorder and an ancestry-matched population genetic reference panel (e.g., 1000 Genomes), which are generally publicly available. In step 1, an existing Bayesian method (e.g., PRScs[12] or SBayesR[11]) is used to compute posterior mean betas on the standardized observed scale with 50% case ascertainment ($p = 0.5$; see Supplementary Note 2); PRScs and SBayesR require slightly different approaches (see Methods). In step 2, the posterior mean betas are transformed to the continuous liability scale

(see Methods and Supplementary Note 3), which are used to construct the PGS. In step 3, BPC requires an estimate of $R^2_{liability}$, the coefficient of determination on the liability scale[7], to derive the distribution of the PGS in cases and controls (see Supplementary Note 4). $R^2_{liability}$ is estimated in an ancestry-matched population reference sample without phenotype information (see Methods). Based on $R^2_{liability}$ and the population prevalence, the expected distribution of PGSs for cases and controls is computed. Lastly, in step 4, the BPC approach uses these distributions and applies Bayes' Theorem to update the prior to the posterior disorder probability based on the individual PGS value.

We compare the BPC approach to one other summary-statistics-based method, introduced in Pain et al.[14]. The approach works as follows. First, the difference in mean PGS between cases and controls is computed based on an estimate of the $R^2$. The $R^2$ is estimated based on the GWAS summary statistics using lassosum[16]. Second, the PGS distribution across cases and controls is divided into quantiles, and third, the disorder probabilities per PGS quantile are assessed based on the prior disorder probability, which gives the predicted disorder probability for individual i. Key differences between the BPC and Pain et al.[14] approach are provided in the Methods.

We also compare the BPC approach to two approaches using phenotype-informed tuning data, BPC-tuned and Logit-tuned. BPC-tuned is identical to BPC, but uses empirical estimates of the distribution of the PGSs in cases and controls derived from the tuning sample with both genotype and phenotype data, instead of deriving them theoretically. The Logit-tuned approach estimates predicted disorder probabilities by fitting a logistic regression model of disease status on PGS in the tuning sample, applying the resulting slope and intercept to PGSs in the testing sample to compute logits, and then transforming these logits using the inverse logit function to disorder probabilities (see Methods for details).

To assess calibration, we compute the Integrated Calibration Index (ICI): the weighted average of the absolute difference between the real and predicted disorder probability[17]. (The real disorder probability is computed using the loess smoothing function in R; thus, the ICI can be intuitively understood as the weighted difference between the calibration curve and the diagonal line in a calibration plot (see **Results: Empirical analysis**). Lower values of the ICI indicate better calibration, and perfect calibration implies an ICI of 0. To assess the

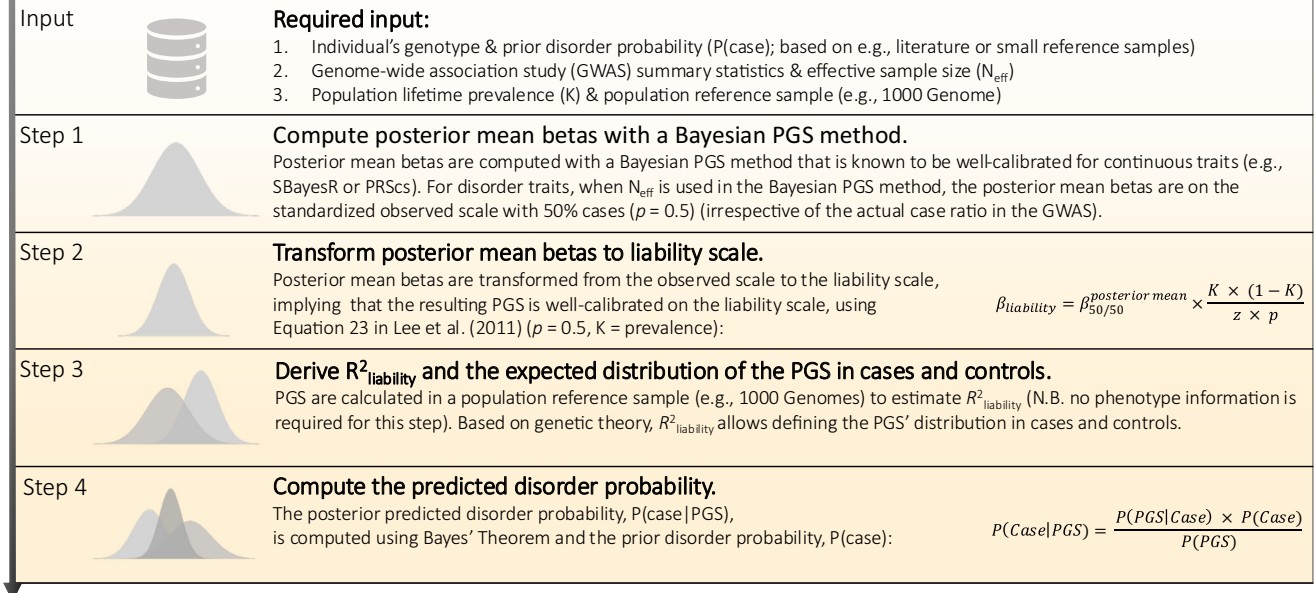

| | |
|---|---|
| Input | **Required input:**<br>1. Individual's genotype & prior disorder probability (P(case); based on e.g., literature or small reference samples)<br>2. Genome-wide association study (GWAS) summary statistics & effective sample size ($N_{eff}$)<br>3. Population lifetime prevalence (K) & population reference sample (e.g., 1000 Genome) |
| Step 1 | **Compute posterior mean betas with a Bayesian PGS method.**<br>Posterior mean betas are computed with a Bayesian PGS method that is known to be well-calibrated for continuous traits (e.g., SBayesR or PRScs). For disorder traits, when $N_{eff}$ is used in the Bayesian PGS method, the posterior mean betas are on the standardized observed scale with 50% cases ($p = 0.5$) (irrespective of the actual case ratio in the GWAS). |
| Step 2 | **Transform posterior mean betas to liability scale.**<br>Posterior mean betas are transformed from the observed scale to the liability scale, implying that the resulting PGS is well-calibrated on the liability scale, using Equation 23 in Lee et al. (2011) ($p = 0.5$, K = prevalence): $\beta_{liability} = \beta^{posterior\ mean}_{50/50} \times \frac{K \times (1 - K)}{z \times p}$ |
| Step 3 | **Derive $R^2_{liability}$ and the expected distribution of the PGS in cases and controls.**<br>PGS are calculated in a population reference sample (e.g., 1000 Genomes) to estimate $R^2_{liability}$ (N.B. no phenotype information is required for this step). Based on genetic theory, $R^2_{liability}$ allows defining the PGS' distribution in cases and controls. |
| Step 4 | **Compute the predicted disorder probability.**<br>The posterior predicted disorder probability, P(case\|PGS), is computed using Bayes' Theorem and the prior disorder probability, P(case): $P(Case\|PGS) = \frac{P(PGS\|Case) \times P(Case)}{P(PGS)}$ |

**Fig. 1 | Overview of the Bayesian polygenic score Probability Conversion (BPC) approach.** The BPC approach transforms an individual's Polygenic Score (PGS) into a well-calibrated disorder probability.

prediction accuracy of the PGSs, we use the Area Under the Curve (AUC) and the $R^2$.

We evaluated the BPC approach in simulations and empirical analyses of nine disorders. In our empirical analyses, we analyzed nine phenotypes based on large training samples of GWAS meta-analyses, namely schizophrenia (SCZ)[18], major depression (MD)[19], breast cancer (BC)[20], coronary artery disease (CAD)[21], inflammatory bowel disease (IBD)[22], multiple sclerosis (MS)[23], prostate cancer (PC)[24], rheumatoid arthritis (RA)[25], and type 2 diabetes (T2D)[26] (see Table 1). We computed the PGSs in three testing samples that were fully independent of the respective training samples: PGC-MD[19], PGC-SCZ[18], and UK Biobank[27] (see Table 1), and we use the 1000 Genomes[28] sample as ancestry-matched population reference sample without phenotype information. The analyses were conducted in individuals of European ancestry, and the tuning approaches were only applied in empirical analyses.

## Simulation analysis

We simulated individual-level data for 1000 SNPs in Linkage Equilibrium based on the liability threshold model[29] (see Supplementary Note 5 for details); we used this simplified simulation setup to limit computational costs (see Methods). We repeated the simulations 100 times for every parameter setting ($R^2_{liability}$: 1%, 5%, 10%, and 15%; population lifetime prevalences: 1% and 15%). We s«imulated three independent samples: a training sample with case-control information used to estimate SNP effects in a GWAS, a population reference sample without case-control information to estimate $R^2_{liability}$ as described above ($N = 503$), and a testing sample with case information to evaluate model performance ($N_{case} = 1000$ and $N_{control} = 1000$). We evaluated the BPC and Pain et al.[14] approaches across all parameter combinations. The BPC approach consistently achieves mean ICI values close to 0 (ranging from mean 0.014 ($\pm$ SE 0.0004) to 0.017 ($\pm$ 0.0006) across $4 \times 2 = 8$ parameter settings), meaning the predicted and observed probabilities agree closely (see Fig. 2).

The Pain et al.[14] approach performs considerably less well (ICI ranging from 0.039 ($\pm$ 0.002) to 0.118 ($\pm$ 0.009) across all parameter settings; see Fig. 2) because it does not distinguish the prior disorder probability (in this case, the testing sample case-control ratio) from the lifetime prevalence in the full population, which overestimates the predicted probabilities and negatively impacts calibration (see **Methods** for details and Supplementary Fig. 1 for a schematic representation). Indeed, the distinction between the BPC and Pain et al.[14] approach is more pronounced when the disorder population lifetime prevalence is low because this increases the difference between the population lifetime prevalence and the prior disorder probability (which is set to 50%). Similarly, larger values of $R^2_{liability}$ exacerbate the overestimates of the Pain et al.[14] approach because it leads to more power to detect the bias (except for $R^2_{liability} = 1\%$; see below). A simple adaptation of the Pain et al.[14] approach to take both the population lifetime prevalence and prior disorder probability into account strongly improves its calibration and removes the negative impact of the low population lifetime prevalence and increasing $R^2_{liability}$ values; nevertheless, the BPC approach continues to achieve lower ICI values (see Supplementary Fig. 2). For low simulated values of $R^2_{liability}$, when the GWAS has little power, the $R^2_{liability}$ values estimated with lassosum in the Pain et al.[14] approach become unstable (see below), leading to an increased ICI. When we adjust the Pain et al.[14] approach to take both the population lifetime prevalence and prior disorder probability into account and compute the variance of a well-calibrated PGS in a population reference sample to estimate $R^2_{liability}$ (instead of lassosum), the difference between both approaches becomes very small (see Supplementary Fig. 3). Nonetheless, the BPC approach achieves slightly better calibration in nearly every condition, because the Pain et al.[14] approach assumes that the variance of the PGS is the same in cases and controls while they are different. The difference becomes

**Table 1 | Phenotype summary**

| Phenotype | Abbreviation | Population lifetime prevalence | GWAS reference | Training sample Effective sample size* ($N_{case}$ / $N_{control}$) | Testing sample Effective sample size* ($N_{case}$ / $N_{control}$) | Testing sample Individual-level dataset |
|---|---|---|---|---|---|---|
| Major Depression | MD[19] | 16.00%[58] | Wray et al.[19] | 133,299** (50,968/96,399) | 25,184*** (12,592/12,592) | PGC-MD[19] |
| Schizophrenia | SCZ | 1.00% | Trubetskoy et al.[18] | 115,996** (48,650/70,612) | 85,340*** (42,670/42,670) | PGC-SCZ[18] |
| Breast Cancer | BC | 12.50% | Zhang et al.[20] | 231,040 (133,384/113,789) | 18,456 (9228/9228) | UKB[27] |
| Coronary Artery Disease | CAD | 3.00% | Nikpay et al.[21] | 129,014 (61,289/126,310) | 20,000 (10,000/10,000) | UKB[27] |
| Inflammatory Bowel Disease | IBD | 1.30% | Liu et al.[22] | 30,273 (12,924/21,770) | 5924 (2962/2962) | UKB[27] |
| Multiple Sclerosis | MS | 0.16% | International Multiple Sclerosis Genetics Consortium (2019)[23] | 35,828 (14,802/26,703) | 2368 (1184/1184) | UKB[27] |
| Prostate Cancer | PC | 12.50% | Schumacher et al.[24] | 125,417 (79,148/61,106) | 7026 (3513/3513) | UKB[27] |
| Rheumatoid Arthritis | RA | 0.50% | Ishigaki et al.[25] | 58,012 (22,350/74,823) | 5076 (2538/2538) | UKB[27] |
| Type 2 Diabetes | T2D | 5.00% | Mahajan et al.[26] | 158,261 (55,005/400,308) | 20,000 (10,000/10,000) | UKB[27] |

*PGC-MD* Major Depressive Disorder Working Group of the Psychiatric Genomics Consortium, *PGC-SCZ* Schizophrenia Working Group of the Psychiatric Genomics Consortium, *UKB* UK Biobank.

\* The effective testing sample size is reported for a testing sample case-control ratio of $P = 0.5$. For analyses with testing sample case-control ratios of $P = 0.25$ and 0.75, cases and controls were down-sampled respectively.

\** The average effective sample size for leave-one-cohort-out GWASs is reported.

\*** The total effective sample size across all cohorts is reported.

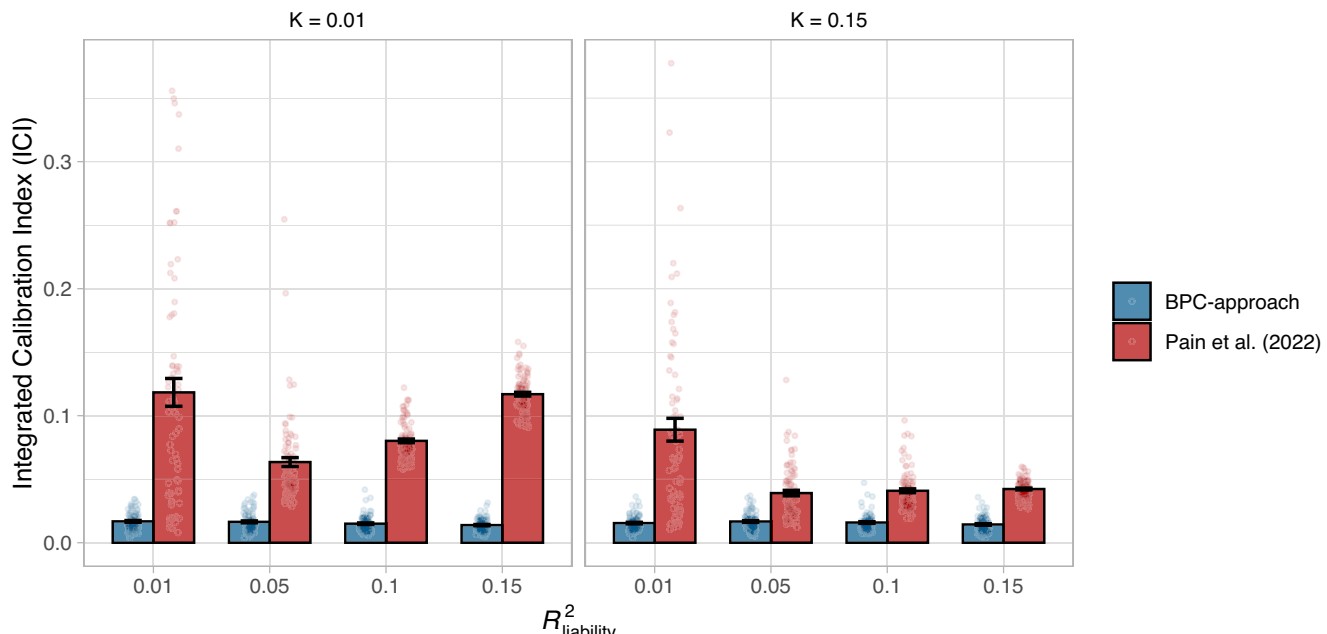

**Fig. 2 | Calibration in simulations.** Calibration of the BPC and the Pain et al.[14] approach was evaluated using the Integrated Calibration Index (ICI) in 100 simulation runs and for combinations of two parameters, the population lifetime prevalence (K), and the explained variance of the PGS on the liability scale ($R^2_{liability}$). The BPC approach achieves low mean ICI values in every condition, while the mean ICI values of the Pain et al.[14] approach are consistently larger. The difference between both approaches becomes larger for conditions with low population lifetime prevalences and large $R^2_{liability}$ values. Error bars represent standard errors and their center represent means.

larger for higher $R^2_{liability}$ values and lower population lifetime prevalences (see Supplementary Fig. 4 and *Methods*).

We conducted several secondary analyses. First, we verified that doubling the number of causal SNPs does not affect these results, and the ICI of the BPC approach remains low (0.016 ± 0.008; $R^2_{liability} = 0.05$ and $K = 0.01$). Second, in addition to the ICI, we used the calibration slope and intercept to evaluate calibration. Again, the BPC approach consistently achieves good calibration (see Supplementary Figs. 5 and 6) and performs better than the Pain et al.[14] approach. Furthermore, the Pain et al. approach consistently overestimates the disorder probabilities, with slopes smaller than one and/or intercepts smaller than zero (see *Methods*). In line with observations made in ref. 17, we show that the ICI is a more stable metric of calibration, especially at small values of $R^2_{liability}$ (see Supplementary Fig. 7).

We also evaluated a linear rescaling approach (see *Methods*). We found that the linear rescaling approach performs reasonably well but worse than the BPC approach because it can result in probabilities larger than 1 and lower than 0. This mostly occurs in conditions where the population lifetime prevalence is low and $R^2_{liability}$ is large. Setting these outlying values to 1 and 0, respectively, negatively impacts calibration (see Supplementary Fig. 8). Therefore, our primary recommendation is to use the BPC approach.

We found that the calibration slopes of untransformed Bayesian PGSs for binary disorder traits deviate from 1 in ascertained samples, even when the case-control ratios in the training and testing sample are both 50% and the PGSs are on the standardized observed scale with 50% case ascertainment. Similarly, the calibration intercepts deviate from 0 (see Supplementary Figs. 9 and 10; the bias is most apparent when the population lifetime prevalence is low and $R^2_{liability}$ is large). This is because the transformation from the liability to the observed scale in ascertained samples is linear for the GWAS results (i.e., betas) used to compute the PGS[30] but non-linear for the coefficient of determination ($R^2$) of the PGS[7] (see Supplementary Fig. 11). As a result, var(PGS$_{observed}$) and $R^2_{observed\ scale}$ are not proportional, and the PGSs can thus not be well-calibrated (see Eq. 2) without a probability

conversion approach. Untransformed PGS do attain accurate calibration when neither the training nor the testing sample case-control ratios differ from the population lifetime prevalence (i.e., random ascertainment), even when the population lifetime prevalence is low ($K = 0.01$) and $R^2_{liability}$ is large (0.15). The PGS's mean calibration slope over 100 simulation runs does not significantly differ from 1 (mean calibration slope = 1.02, s.e.m. = 0.02). We note that the untransformed Bayesian PGSs are centered around zero and cannot be evaluated with the ICI[17].

The BPC approach assumes that the PGSs are normally distributed in cases and controls. We verified that this assumption holds for all parameters in our simulations and that significant deviations are only observed at current unrealistically large values of $R^2_{liability}$ (=0.6; see Supplementary Fig. 12). A second assumption is that the liability conversion of the PGS is successful. We verified that regressing the liability scores on the PGSs (based on Bpred, a version of LDPred that assumes linkage equilibrium[13]) in a population reference sample leads to slopes and intercepts that are, on average, 1 and 0, respectively (see Supplementary Fig. 13).

Lastly, we investigated the distribution of $\frac{P(PGS_i|D_i = case)}{P(PGS_i)}$ (see Eq. 3) to test how strongly the posterior predicted disorder probabilities depend on the prior ($P(D_i = case)$). If the probabilities are determined mainly by the prior, the distribution is expected to vary closely around 1. We find that the distributions vary markedly around 1 for most realistic simulation conditions (e.g., S.D. = 0.3 for $K = 0.01$, $R^2_{liability} = 0.05$, and prior = 0.50), except when the $R^2_{liability}$ is very low, the population prevalence is high, and the prior is very high (i.e., S.D. = 0.05 for $R^2_{liability} = 0.01$, $K = 0.15$, and prior = 0.75) (Supplementary Fig. 14).

### Empirical analysis

To further evaluate the performance of the BPC approach, we applied it to nine phenotypes across nine training samples (SCZ[18], MD[19], BC[20], CAD[21], IBD[22], MS[23], PC[24], RA[25], and T2D[26]) and three testing samples (i.e., UK Biobank[27], PGC-SCZ[18], PGC-MD[19]; see *Methods* and Table 1 for a summary). We ascertained cases and controls for each phenotype such

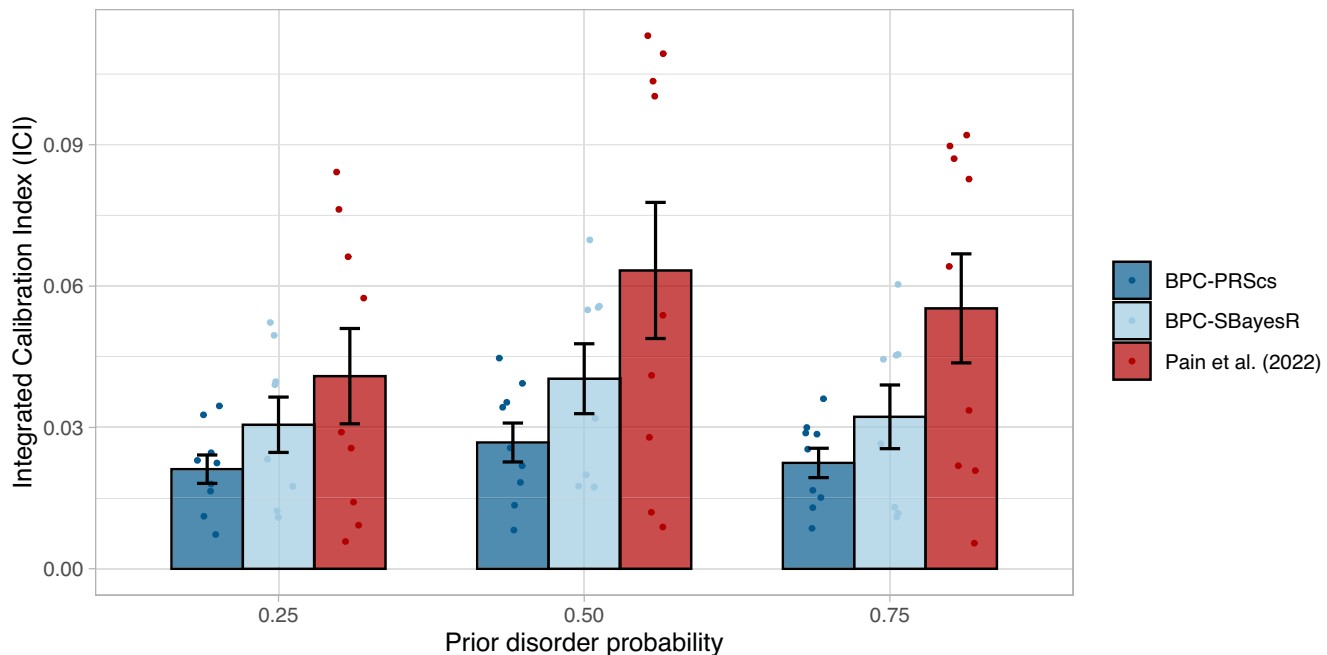

**Fig. 3 | Calibration in empirical analyses of nine disorders.** Calibration of the BPC and the Pain et al.[14] approach was evaluated using the Integrated Calibration Index (ICI) for nine disorders, while varying the prior disorder probability. The BPC approach was applied using two Bayesian PGS methods, PRScs (BPC-PRScs) and SBayesR (BPC-SBayesR). The BPC-PRScs approach achieves the lowest mean ICI values across all prior disorder probabilities. BPC-SBayesR shows one fewer data points, as it did not converge for prostate cancer. Numerical values are presented in Supplementary Data 1. Error bars represent standard errors and their center represent means.

that the testing sample case-control ratios were 0.25, 0.5, and 0.75, thus testing the calibration of the BPC approach across a range of prior disorder probabilities. We performed similar comparisons as in the simulations with the addition of two applications of the BPC approach, one using PRScs[12] (BPC-PRScs) and one using SBayesR[11] (BPC-SBayesR) to compute posterior mean betas (see Fig. 1 and *Methods*). We note that for SBayesR, the results did not converge for prostate cancer and therefore depict one fewer data point. Results are reported in Fig. 3 and Supplementary Data 1. Averaged across all prior disorder probabilities, BPC-PRScs achieves the lowest mean ICI value of 0.024 (±0.002), followed by BPC-SBayesR with 0.034 (±0.004). The Pain et al.[14] approach has the largest mean ICI value of 0.053 (±0.007). The BPC-PRScs approach consistently achieves the lowest mean ICI values across all prior disorder probabilities. We note the Pain et al.[14] approach can be used with both PRScs and SBayesR. While the presented results are based on PRScs, using SBayesR yields comparable results (see Supplementary Fig. 15 and Supplementary Data 1). The observation that the BPC approach produces well-calibrated predicted disorder probabilities suggests that the PGSs are also well-calibrated on the unobserved liability scale.

When focusing in detail on the calibration plots with a prior disorder probability of 50%, BPC-PRScs shows better calibration than the Pain et al.[14] approach for every trait, except Type 2 Diabetes (see Fig. 4 and Supplementary Data 1). The Pain et al.[14] approach tends to overestimate the probabilities for many traits, as can be seen by the right shift of the histograms and calibration lines. This is particularly true for traits with low population lifetime prevalence and large $R^2_{liability}$ values, such as rare auto-immune disorders (i.e., Inflammatory Bowel Disorder, Multiple Sclerosis, and Rheumatoid Arthritis) and Prostate Cancer, which is in line with our theoretical expectations (see *Methods* and Supplementary Fig. 1 for a schematic representation).

We performed secondary analyses yielding the following eight conclusions. First, comparing the calibration plots of BPC-PRScs with BPC-SBayesR, the latter makes correct predictions on average but is less well-calibrated for low and high values of the predicted disorder

probabilities (see Supplementary Fig. 16 and Supplementary Data 1). Second, misspecification of the effective sample size by a factor of 0.5 and 2 negatively impacts calibration for BPC-PRScs, while it does not affect the calibration of the Pain et al.[14] approach (see Supplementary Fig. 17 and Supplementary Data 2) as it involves a scaling step after the posterior mean betas have been computed. We note the BPC approach still has lower median ICI values than the Pain et al.[14] approach. BPC-SBayesR seems generally more robust to misspecification of the effective sample size, except for Coronary Artery Disease, which suffers extreme miscalibration when $N_{eff}$ is multiplied by 2. Third, misspecification of the prior impacts calibration because it shifts the mean of the predicted disorder probabilities. A mismatch of 0.25 between the true and assumed prior leads to an average increase of 0.21 (s.e.m. 0.02) in the ICI (see Supplementary Fig. 18). However, given that the BPC approach is well-calibrated under a range of correctly specified priors, the change of the posterior predicted disorder probability relative to the prior remains informative as it makes the diagnosis less or more likely compared to the prior expectation. In practice, the prior can be estimated from small reference samples, literature, or prior elicitation (see *Discussion* for more information). Fourth, including the MHC region strongly and negatively impacts calibration for the auto-immune disorders Multiple Sclerosis and Rheumatoid Arthritis for BPC-PRScs and Pain et al.[14] (but not BPC-SBayesR; This is because SBayesR's reference sample excludes most of the MHC region; see Supplementary Fig. 19 and Supplementary Data 3). Fifth, reducing the INFO filter from 0.9 to 0.3 and the minor allele frequency filter from 10% to 1% (as in ref. 31) yields comparable average ICI values (except for Coronary Artery Disease and BPC-SBayesR; see Supplementary Fig. 20 and Supplementary Data 4). Sixth, evaluating calibration with the slope and intercept from a linear regression of the phenotype on the predicted disorder probabilities also shows that BPC-PRScs is best calibrated overall (see Supplementary Figs. 21 and 22, and Supplementary Data 5). Seventh, we tested and confirmed that the BPC's assumption of normally distributed PGSs in cases and controls holds for all analyzed phenotypes (see Supplementary Fig. 23). Eighth, we investigated

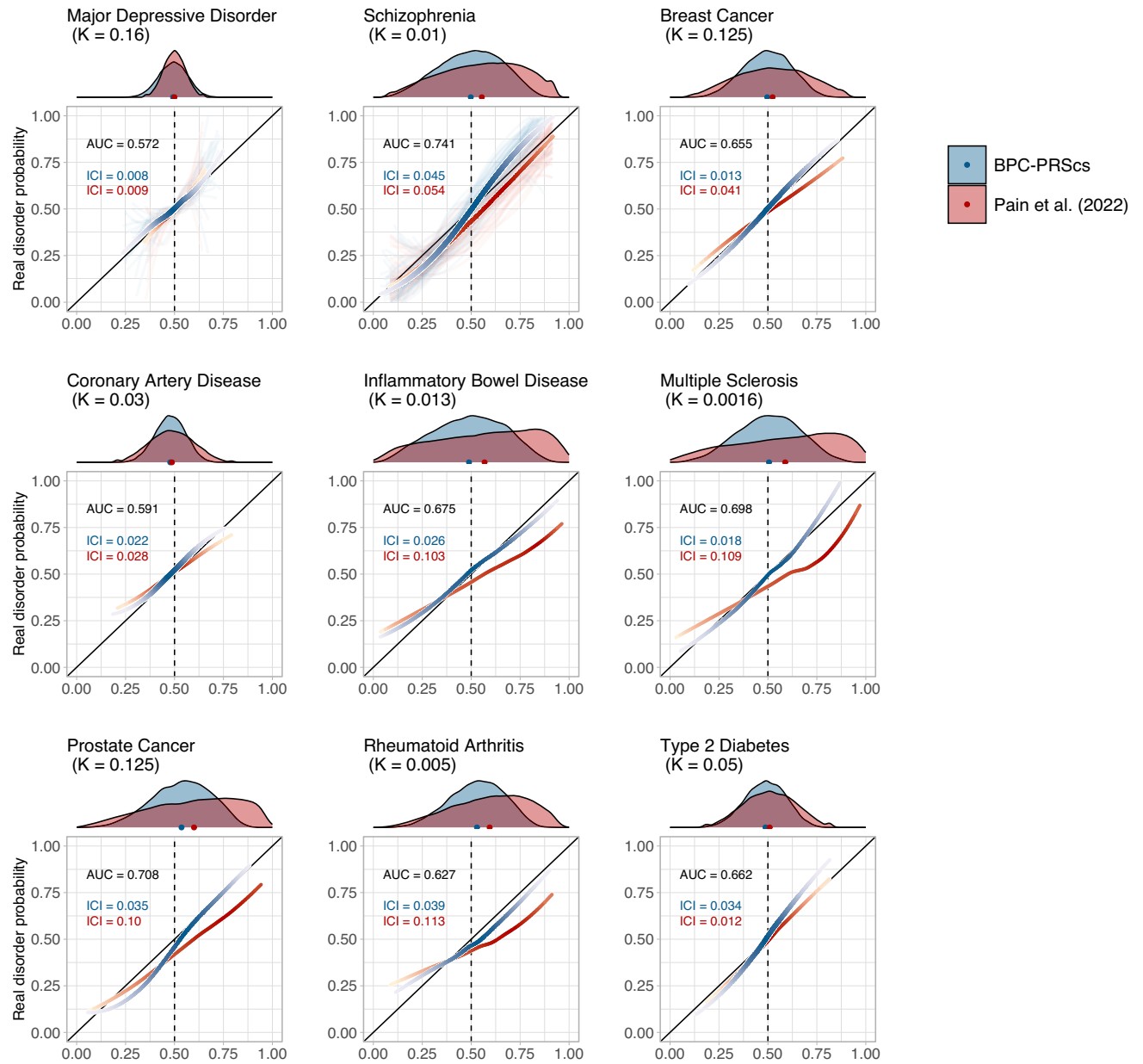

**Fig. 4 | Disorder-specific calibration curves in empirical analyses of nine disorders.** Calibration of the BPC and the Pain et al.[14] approach was evaluated using the Integrated Calibration Index (ICI) for nine disorders, each with a prior disorder probability of 0.5 (see Table 1 for an overview of the case/control testing sample sizes). The prior disorder probability was set to 0.5, as opposed to the lifetime prevalence in the general population (K), to emulate the higher risk of help-seeking individuals in clinical settings. Histograms at the top of the plots depict the distribution of the predicted disorder probabilities, and the dots at the base of the histograms depict the mean predicted probability. The lines were drawn with a loess smoothing function, and their transparency follows the density of the histogram to show which parts of the distribution carry the most weight in the calculation of the ICI. For major depression and schizophrenia, 62 and 22 cohorts, respectively, were available for analysis and therefore depict thin, light-colored, and transparent lines for individual cohorts. In contrast, the thicker and darker lines depict results when data from all cohorts are concatenated. The disorder population lifetime prevalence (K) is reported. The Area Under the receiver operator Curve (AUC) is the same for both approaches because the transformations do not change the ranking of individual PGSs, and both approaches use the same PGS inputs. The BPC-PRScs approach achieves lower ICI values for eight out of nine disorders. The Pain et al.[14] approach tends to overestimate the predicted disorder probabilities, as seen by the right shift of the histograms and the dots. Numerical values are presented in Supplementary Data 1. Calibration curves for BPC-SBayesR are presented in Supplementary Fig. 16.

the distribution of $\frac{P(\mathrm{PGS}_i | D_i = \mathrm{case})}{P(\mathrm{PGS}_i)}$ and found it to vary considerably around one (e.g., S.D. = 0.29 for schizophrenia when the prior = 0.50), showing that the predicted disorder probabilities are not solely determined by the prior (see Supplementary Fig. 24).

In contrast to simulations (see Supplementary Figs. 9 and 10), the untransformed Bayesian PGSs do not show strongly miscalibrated slopes and intercepts (see Supplementary Figs. 25 and 26), likely due to

the variance of estimates of the calibration slopes in combination with much fewer observations in empirical data (i.e., 9) than in simulations (100 simulation runs for 8 parametrizations). Our findings align with the previous observation that the calibration slope is very sensitive to miscalibration in small parts of the data and that the ICI is more robust and preferred as a metric for calibration[17]. Because untransformed Bayesian PGSs are centered around 0 and do not range from 0 to 1,

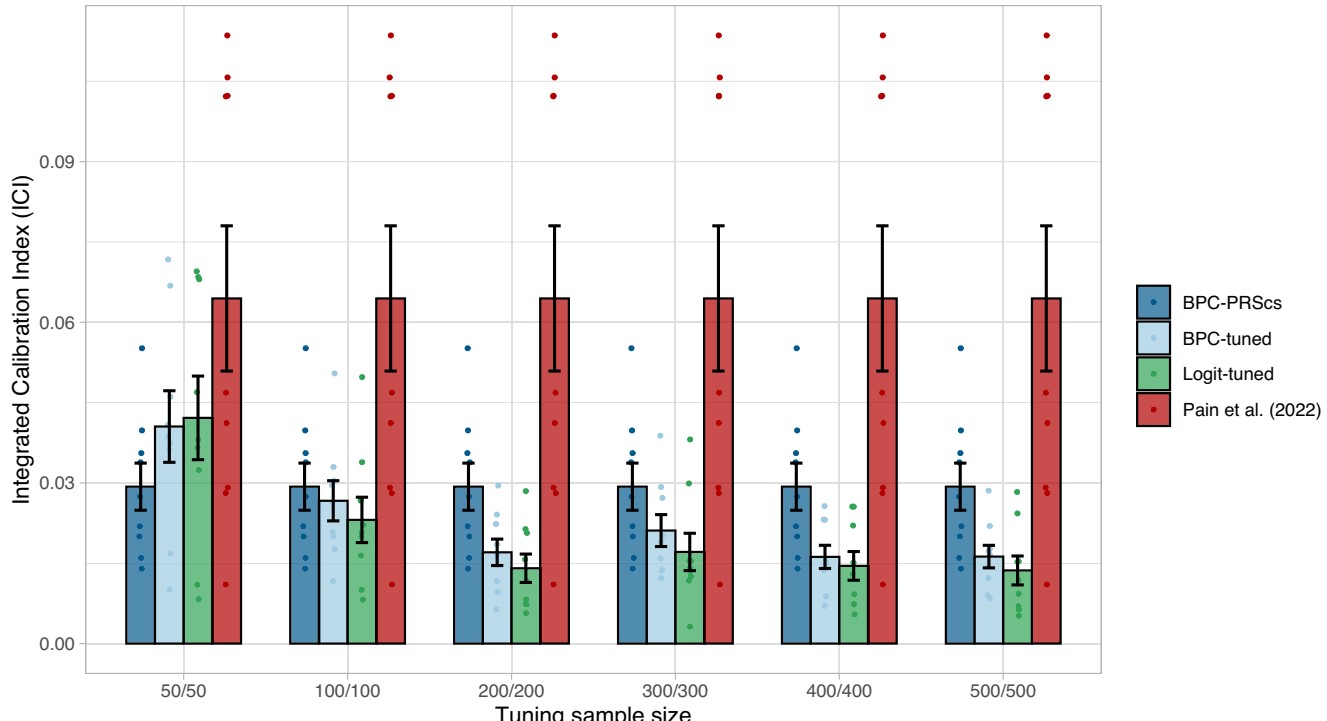

**Fig. 5 | Calibration of tuning approaches in empirical analyses of nine disorders.** Calibration of the BPC-PRScs, BPC-tuned, Logit-tuned and the Pain et al.[14] approach was evaluated using the Integrated Calibration Index (ICI) for nine disorders. BPC-tuned and Logit-tuned use a tuning sample that includes genotype and phenotype data, whereas BPC-PRScs and Pain et al.[14] do not require an additional independent tuning sample. Tuning sample sizes are presented as ($N_{case}/N_{control}$). Error bars represent standard errors and their center represent means.

they cannot be evaluated with the ICI and cannot be interpreted as predicted disorder probabilities.

### Comparing to calibration of tuning approaches

The BPC approach does not require tuning samples to estimate predicted disorder probabilities. However, to benchmark the BPC approach, we compared it to other approaches that utilize such tuning samples that include genotype and phenotype data (see **Methods**). The calibration of the BPC approach is similar to the tuning approaches when the tuning samples are smaller than 200 cases and 200 controls (see Fig. 5), while the area under the ROC curve (AUC) does not differ between these approaches (see Supplementary Fig. 27). For larger tuning sample sizes, the tuning approaches have an ICI that is approximately 0.015 smaller. However, we consider BPC's calibration (ICI < 0.03) satisfactory, such that the benefit of not requiring a tuning sample outweighs the improved calibration of the tuning approaches.

### Estimation of variance explained ($R^2_{liability}$)

The BPC approach depends on a valid estimate of $R^2_{liability}$. We compute the variance of a well-calibrated PGS in a population reference sample without the need for phenotype data (see *Methods*). This leads to estimates that are very close to the observed values from linear regression[7] in a sample with both pheno- and genotype data in simulations (mean absolute difference ranges from 0.009 to 0.011; see Fig. 6a) and in empirical data (mean absolute difference = 0.02; Fig. 6b). This suggests that the PGSs are well-calibrated on the unobserved liability scale. The Pain et al.[14] approach uses lassosum[16], which leads to estimates that are slightly misspecified in simulations (mean absolute difference ranges from 0.058 to 0.088) and in empirical data (mean absolute difference = 0.05).

## Discussion

We developed the BPC approach to transform PGSs to absolute risk values, which yields predicted disorder probabilities that may be clinically useful for single individuals. Based on Bayesian PGS methods, it requires only minimal input, namely GWAS summary statistics, a single individual's genome-wide genotype data and prior disorder probability, and an estimate of the disorder's population lifetime prevalence. We verified in simulations and empirical analyses of nine disorders that the BPC approach achieves good calibration across a range of prior disorder probabilities, meaning the predicted and real disorder probabilities closely align. The BPC approach depends on a valid estimate of $R^2_{liability}$, which we compute by estimating the variance of a well-calibrated PGS in a population reference sample without the need for phenotype data, and verify that the estimates are close to empirically calculated values in case-control data.

We compared the BPC approach to a recently published approach in Pain et al.[14], and showed that it achieves lower ICI values in every simulation condition and for eight out of nine tested disorders in empirical analyses. This is partly because the Pain et al.[14] approach overestimates the predicted disorder probabilities whenever the prior disorder probability exceeds the population lifetime prevalence. We also compared the BPC approach to methods requiring tuning data[10]. We found that for larger tuning sample sizes of more than 200 cases and controls, the tuning approaches have an ICI that is approximately 0.015 smaller. However, we consider BPC's calibration (ICI < 0.03) satisfactory, such that the benefit of not requiring a tuning sample outweighs the improved calibration of the tuning approaches.

In clinical settings where a single individual may be considered, the prior disorder probability, which can be interpreted as the case-control ratio in a hypothetical testing sample to which that individual belongs, can be approximated in several ways. It may be estimated using a small external reference sample to obtain a data-informed

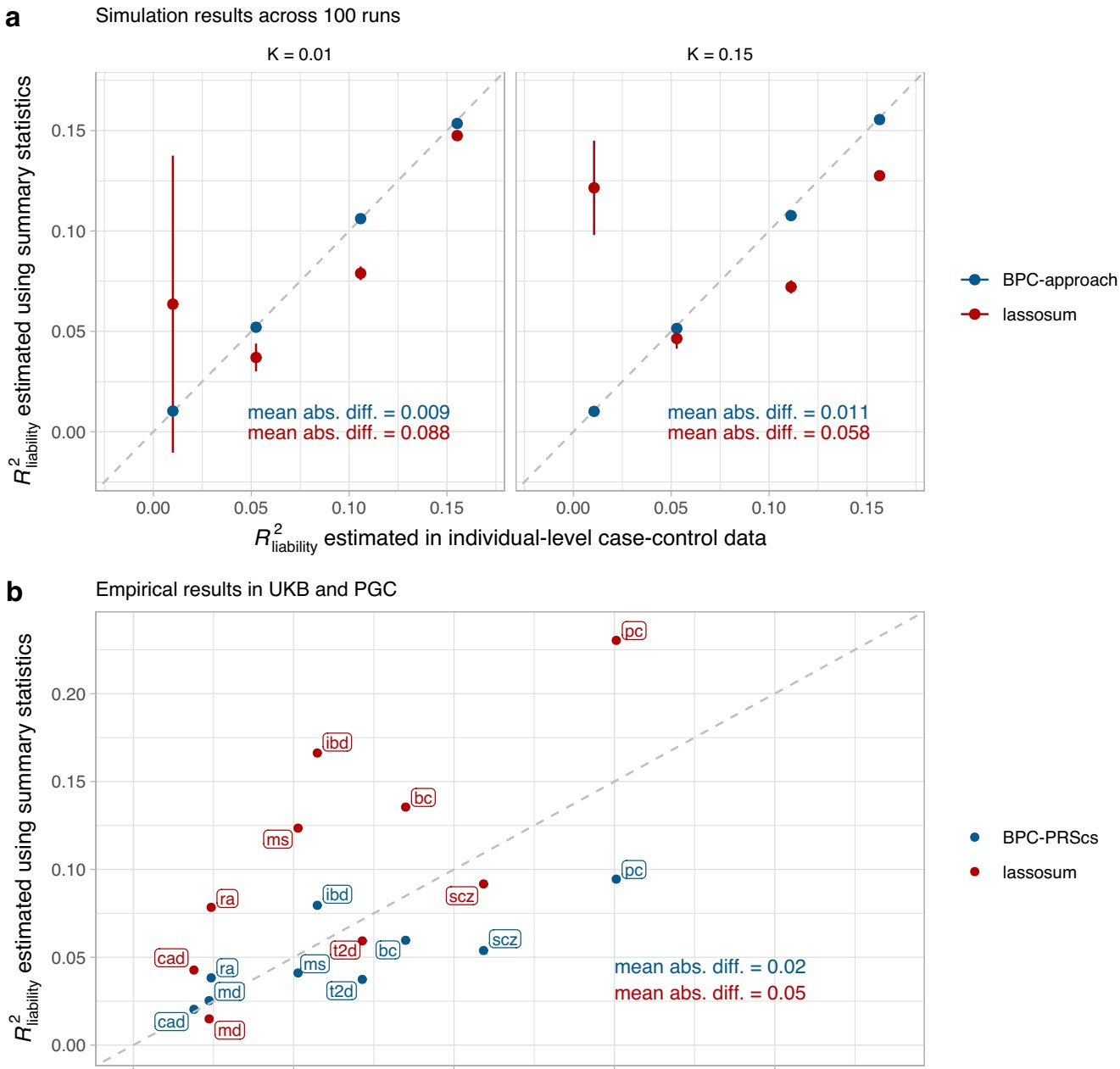

**Fig. 6 | $R^2_{\text{liability}}$ estimates in simulations and empirical analyses of nine disorders. a** Simulation results of estimating $R^2_{\text{liability}}$ using the BPC approach and lassosum (as used by Pain et al.[14]), both of which do not require disorder-specific individual-level genotype and phenotype data. The x-axis depicts $R^2_{\text{liability}}$ estimated by regressing disorder status on the Bayesian PGS in individual-level data in the testing sample[7]. Error bars depict standard errors for 100 simulation runs. The gray dashed line depicts the identity line when y = x. The BPC approach achieves mean estimates that are closer to the regression results in the testing sample in every simulation condition. mean abs. diff. = mean absolute difference of $R^2_{\text{liability}}$ estimates using summary statistics and individual-level case-control data. **b** Empirical results in the UKB and PGC of estimating $R^2_{\text{liability}}$ using the BPC-PRScs approach and lassosum. The BPC-PRScs approach achieves estimates that are closer to the regression results in the testing sample on average (mean absolute difference of 0.02 vs. 0.05).

prior, such as context-specific prevalence estimates of individuals seeking health care for a specific disorder in a given hospital. Such a reference sample does not require genotype data and may be smaller than those required for the tuning approaches. Alternatively, such context-specific prevalence estimates may also be obtained from the literature[32]. The context may refer to any variable that modifies a disorder's prevalence, such as age or sex[33]. When no data is available to estimate the prior, *prior elicitation*[15] may be used, where a clinician (or

a panel of clinicians) provides a subjective estimate of the prior. Generally, the lifetime risk for help-seeking individuals is expected to be higher than for individuals from the general population (where lifetime risk = K). As such, the prior will often be higher than K. When considerable uncertainty about the prior exists, a range of priors may be used to obtain a range of posterior disorder probabilities.

There are several limitations to this study. First, because most GWASs are based on individuals from European populations, the

calibration of the BPC approach for individuals from non-European populations is unknown but may be negatively affected, as is the accuracy of risk predictions[34,35]. However, as long as the GWAS population matches that of the individual, the BPC approach is expected to be well-calibrated. Future studies are needed to develop methods to obtain well-calibrated predictions for individuals from non-European populations. Second, we performed simulations without LD, which may be perceived as a limitation. However, we note that the results from our simulation and empirical analyses were concordant, suggesting that our simplified simulation setup was appropriate. Third, the potential for clinical utility of polygenic prediction (and thereby the BPC approach) strongly depends on the magnitude of the PGS's $R^2_{\text{liability}}$, which is currently prohibitively small for most traits. However, there are traits, such as coronary artery disease[36–38], type 2 diabetes[39,40], breast cancer[4,41,42], chronic obstructive pulmonary disease[5], and prostate cancer[6,43], for which current PGSs may already be sufficiently powered to find clinical application and be economically effective. Moreover, as GWAS sample sizes grow, the PGS's $R^2_{\text{liability}}$ is expected to approach the disorder's $h^2_{\text{SNP}}$, and therefore, their clinical applicability will become more likely. Fourth, the calibration of the predicted disorder probabilities depends on a correct estimate of the prior. While we showed that misspecification of the prior negatively impacts calibration, we also showed that the BPC approach is well-calibrated across a range of correctly specified priors and that, therefore, the change of the posterior predicted disorder probability relative to the prior remains informative. Irrespectively, striving for the best possible prior disorder probabilities in practice is important and provides an important direction for future research. Fifth, the BPC approach can only be applied to polygenic traits with normally distributed PGSs in cases and controls. While we show that this assumption holds in our simulation and empirical analyses (Supplementary Fig. 12), its violation due to outlying common, very large-effect variants can negatively impact calibration, such as APOE for Alzheimer's Disease[44], and should be removed prior to the application of the BPC approach. Integrating prediction based on rare variants with large effects with polygenic prediction is an important direction for future research. Sixth, while variables that are not correlated to the PGS (e.g., sex, age) can easily be used to adjust the prior, variables that are correlated to the PGS (e.g., family history[45–48]) cannot currently be incorporated into the BPC approach because, in this case, the prior cannot be adjusted independently without modifying the $R^2_{\text{liability}}$. Extending the BPC approach to include variables correlated to the PGS is an important direction for future research. Seventh, the BPC outcome is presented as a fixed lifetime probability. Extending the BPC approach to model the decline in risk in the years following the assessment in which the disorder has not manifested is an important direction for future research.

In conclusion, the BPC approach provides an effective tool to compute well-calibrated predicted disorder probabilities based on PGSs.

## Methods
### Bayesian polygenic score Probability Conversion (BPC) approach
We developed the BPC approach to achieve calibration for binary disorder traits in ascertained samples, using the existing Bayesian Polygenic Score (PGS) methods PRScs[12] and SBayesR[11]. The BPC approach follows four steps (see Fig. 1).

First, the BPC approach requires as input an individual's genotype data and prior disorder probability. The prior can be based on context-specific prevalence estimates from published literature[32], small reference samples, or prior elicitation (see *Discussion* for a detailed discussion on approaches to set the prior). For convenience, we mostly report results for a prior of 0.50. Second, the BPC approach requires the GWAS summary statistics (training sample) and the effective sample size ($N_{\text{eff}}$, see Supplementary Note 1) of the training sample

(i.e., the sum of $N_{\text{eff}}$ of all cohorts contributing to the meta-analysis[49]). The GWAS betas are assumed to be age-independent. Third, the population lifetime prevalence of the disorder of interest and an ancestry-matched population reference sample (e.g., 1000 G) are required. No tuning sample with both genotype and phenotype data is required. We note that instead of an individual-level population reference sample, summary-level LD and allele frequency information could, in principle, be used as well. It is important to use the same set of SNPs across the training sample, reference sample, and the individual's genotype data to ensure optimal prediction accuracy and well-calibrated BPC predictions.

The BPC approach requires the posterior mean betas to be on the standardized observed scale with 50% case ascertainment ($p = 0.5$). For PRScs, this is achieved by simply using $N_{\text{eff}}$ (i.e. the effective sample size)[49] as input because PRScs is based on the GWAS Z-scores, noting that $\beta_{50/50} = z/\sqrt{N_{\text{eff}}}$.[50] (see Supplementary Note 2). (We note that, as long as $N_{\text{eff}}$ is used, the proportion of cases in the discovery GWAS can have different values from 50%.) In contrast, SBayesR is based on the GWAS effect sizes (typically on the log-odds scale), which first need to be transformed to $\beta_{50/50} = z/\sqrt{N_{\text{eff}}}$ before applying SBayesR, while also setting $N_{\text{eff}}$ as sample size.

The posterior mean betas are transformed from the standardized observed scale with 50% case ascertainment to the continuous liability scale ($\beta_{\text{liability}}$)[30] (see Supplementary Note 3):

$$\beta_{\text{liability}} = \beta_{50/50}^{\text{posterior mean}} \times \frac{K \times (1 - K)}{z \times p} \tag{1}$$

where $K$ denotes the disorder population lifetime prevalence and $z$ is the height of the standard normal probability density function at a threshold corresponding to $K$[30]. Subsequently, a PGS is constructed using $\beta_{\text{liability}}$ and an individual's genotype data.

To define the standard normal probability density function of the PGS in both cases and controls, an estimate of $R^2_{\text{liability}}$, the coefficient of determination on the liability scale[7], is required. When a PGS is well-calibrated for a standardized phenotype with variance 1 (here the liability[51]), the variance of the PGS equals the variance explained by the PGS in the phenotype:

$$R^2_{\text{liability}} = \frac{\text{var}(\text{slope} \times \text{PGS}_{\text{liability}})}{\text{var}(\text{liability})} = \frac{\text{var}(1 \times \text{PGS}_{\text{liability}})}{1} = \text{var}(\text{PGS}_{\text{liability}}) \tag{2}$$

where *slope* refers to the regression of the liability on $PGS_{\text{liability}}$ (which is equal to 1 due to the PGS being well-calibrated). Thus, $R^2_{\text{liability}}$ can be estimated by computing $\text{var}(PGS_{\text{liability}})$ in an ancestry-matched population reference sample without the need for phenotype data. Given $R^2_{\text{liability}}$, the expected mean and variance of the PGS can be estimated in cases and in controls using normal theory[52,53] (see Supplementary Note 4). Thus, the expected conditional probabilities $P(\text{PGS}_i|D_i = \text{case})$ and $P(\text{PGS}_i|D_i = \text{control})$ can be estimated for every individual $i$ with PGS value $\text{PGS}_i$ and disease status $D_i$.

Finally, we use Bayes' theorem to update the prior disorder probability to the posterior probability:

$$P(D_i = \text{case}|\text{PGS}_i) = \frac{P(\text{PGS}_i|D_i = \text{case}) \times P(D_i = \text{case})}{P(\text{PGS}_i)} \tag{3}$$

where $P(D_i = \text{case})$ is the prior disorder probability for individual $i$, $P(\text{PGS}_i|D_i = \text{case})$ is the conditional probability, and $P(\text{PGS}_i)$ is the normalization factor corresponding to $P$. Thus, the BPC approach provides predicted disorder probabilities for individuals based on GWAS summary statistics, individual genotype data, and a prior disorder probability. (See *Code Availability* for R code to implement the BPC approach.). We note the prior disorder probability can be

specified flexibly and does not depend on the case ratio in the training GWAS sample (see *Discussion* for a detailed discussion on how to set the prior).

## Alternative approaches to obtain disorder probabilities from PGS

The BPC approach transforms a single individual's genotype data to the predicted disorder probability based on only publicly available data without requiring tuning samples that include both pheno- and genotype data, making it practical in its application. We are aware of only one other published approach that computes disorder probabilities only based on publicly available data, introduced in Pain et al.[14]. In addition, we describe the linear rescaling approach, an unpublished alternative to the BPC approach.

Briefly, the approach of Pain et al.[14] works as follows. First, the difference in mean PGS between cases and controls is computed based on an estimate of the $R^2$ (which is transformed to the AUC[54,55]), assuming the PGS have the same variance in cases and controls (scaled to 1). The $R^2$ is estimated based on the GWAS summary statistics using lassosum[16]. Second, the PGS distribution across cases and controls is divided into quantiles, and third, the disorder probabilities per PGS quantile are assessed based on the testing sample's case-control ratio (i.e. the prior disorder probability). For individual $i$, the predicted disorder probability follows by finding which quantile contains its PGS Z-value (standardized based on the distribution of the PGS in 1000 Genomes).

The approach of Pain et al.[14] differs in three important ways from the BPC approach. First, it implicitly assumes that the variance and the mean of the PGS in the full population are the same as in the target sample. However, if the target sample is over-ascertained for cases, the variance and the mean are larger than in the full population (see Fig. S1). As such, PGS Z-values based on the full population (i.e., 1000 Genomes) will overestimate the PGS Z-values in the ascertained target sample and, consequently, also the predicted disorder probabilities. Second, Pain et al.[14] suggest using lassosum[16] to estimate the $R^2$ from summary statistics, while the BPC approach achieves this by estimating the variance of a well-calibrated PGS in a population reference sample. Third, the Pain et al.[14] approach assumes var(PGS|case) = var(PGS|control), while the BPC approach models more precisely the fact that var(PGS|case) < var(PGS|control), which has the most impact for disorders with low population lifetime prevalence (K) and large $R^2_{\text{liability}}$ values (see Results & Supplementary Table 1 for a summary of these differences).

We developed an alternative approach, the linear rescaling approach, to obtain well-calibrated predicted disorder probabilities, that does not apply Bayes' Theorem but a linear rescaling of the $PGS_{\text{liability}}$ instead. The linear rescaling approach follows steps 1-3 of the BPC approach described above and in Fig. 1. Subsequently, the expected variance of the $PGS_{\text{liability}}$ in the ascertained sample, var($PGS_{\text{liability}}$|ascertained sample), is computed based on the prior disorder probability (i.e., the case-control ratio in the testing sample, $P$(case)) and the distribution of $PGS_{\text{liability}}$ in cases and controls. Next, the PGS is scaled to PGS′ with the property that $var\left(PGS'|\text{ascertained sample}\right) = R^2_{\text{observed}}$ in the ascertained sample ($R^2_{\text{observed}}$ is computed based on $R^2_{\text{liability}}$ and the transformation introduced in Lee et al.[7]), resulting in PGS′ that is well-calibrated on the standardized observed scale (see Eq. 2). Lastly, we scale the PGS′ (which is based on a standardized phenotype) to the observed scale with cases coded 1 and controls 0, $PGS_{0-1\text{scale}} = PGS'*\sqrt{P(\text{case})x(1 - P(\text{case}))} + P(\text{case})$, resulting in PGSs that represent the predicted disorder probability. We note the linear rescaling approach can lead to predicted disorder probabilities that are larger than 1 and smaller than 0, which we truncate to 1 and 0 before evaluating its calibration.

## Approaches using tuning samples

We developed an alternative BPC-tuned approach that is conceptually similar to the standard BPC approach outlined above. Instead of deriving them theoretically, it uses empirical estimates of the variances and means of the PGS in cases and controls derived from a tuning sample with both genotype and phenotype data. As such, the BPC-tuned approach skips steps 1 and 2 described above and in Fig. 1.

The Logit-tuned approach, as applied in ref. 10 computes predicted disorder probabilities in three steps. First, the slope and intercept are estimated from a logistic regression model in the tuning sample: $D \sim PGS$, where $D \in \{0, 1\}$ is a vector of binary disease status. Second, the PGSs in the testing sample are used to compute logit($\hat{D}$): $PGS*slope + intercept$. Third, the predicted disorder probabilities are computed as the inverse logit transformation of $\hat{D}$ : $P\left(D_i = \text{case}|PGS_i\right) = \frac{e^{\hat{D}}}{1 + e^{\hat{D}}}$.

## Untransformed PGS

We also evaluated the calibration of untransformed PGSs. These are constructed using the posterior mean betas of step 1 (see Fig. 1), which are on the standardized observed scale with 50% case ascertainment when $N_{\text{eff}}$ is used as input in the Bayesian PGS methods. The resulting PGSs are centered around zero and cannot be interpreted as disorder probabilities.

## Metrics of performance

To assess calibration, we compute the Integrated Calibration Index (ICI): the weighted average of the absolute difference between the real and predicted disorder probability[17]. (The real disorder probability is computed using the loess smoothing function in R; thus, the ICI can be intuitively understood as the weighted difference between the calibration curve and the diagonal line in a calibration plot (see *Results*)). Lower values of the ICI indicate better calibration and perfect calibration implies ICI = 0.

The calibration slope is another metric to assess calibration that is often used in the literature[11-13], which refers to the slope from a linear regression of the phenotype of interest on the PGS. If the slope equals 1 and the intercept 0, the predictor is said to be well-calibrated. A downside of this metric is that a PGS with values outside the range of 0 and 1 can still have a calibration slope of 1, and the ICI has been proposed as a superior metric because the ICI is robust to sparse subregions of poor calibration[17]. Typically, untransformed Bayesian PGSs are centered around 0, and while they may have a calibration slope of 1, they cannot be interpreted as disorder probabilities and cannot be evaluated with the ICI.

To assess the prediction accuracy of the PGSs, we use the Area Under the Curve (AUC) and the coefficient of determination ($R^2$) (we note the AUC and $R^2$ can be transformed into one another[7]).

## Simulation analysis

We simulated individual-level data for 1000 SNPs in Linkage Equilibrium based on the liability threshold model[29] (see Supplementary Note 5 for details). We simulated a relatively small number of SNPs (M) because this allows the simulation of smaller training sample sizes (N), which reduces the computational cost. The PGS's $R^2$ primarily depends on $\frac{M}{N}$, such that simulations at reduced values of both M and N are appropriate[13]. To further reduce the computational cost, we did not simulate Linkage Disequilibrium (LD), which has no impact on the scale of the PGS as it aggregates all SNP effects into a single score. We repeated the simulations 100 times for eight different parameter settings where we varied the power of the training sample and thereby the coefficient of determination ($R^2$) of the PGS ($R^2_{\text{liability}} = \{0.01, 0.05, 0.10, 0.15\}$), as well as the disorder population lifetime prevalence ($K = \{0.01, 0.15\}$). The disorder's SNP-based heritability was set to 0.2. We simulated three independent samples: a training sample with case-control information used to estimate SNP effects with a GWAS (varying N; see

below), a population reference sample without case-control information to estimate $R^2_{\text{liability}}$ as described above ($N = 503$), and a testing sample with case control-information to evaluate model performance ($N_{\text{case}} = 1000$ and $N_{\text{control}} = 1000$). To achieve the desired $R^2_{\text{liability}}$ in the testing sample, we approximated the required sample size of the training sample using the avengeme package in R[56] (e.g. $N_{\text{training}} = 2759$ when $R^2_{\text{liability}} = 0.1$ and $K = 0.01$). We computed the posterior mean betas using Bpred, the version of LDPred that assumes linkage equilibrium[13], with GWAS betas on the standardized observed scale with 50% case ascertainment and therefore used $N_{\text{eff}}$ as input. We applied the BPC approach to estimate predicted disorder probabilities and compared it to the existing approach introduced in Pain et al.[14].

### Empirical analysis

We analyzed nine phenotypes based on large training samples of GWAS meta-analyses, namely schizophrenia (SCZ)[18], major depression (MD)[19], breast cancer (BC)[20], coronary artery disease (CAD; we note that 23% of the training sample included individuals from non-European populations)[21], inflammatory bowel disease (IBD)[22], multiple sclerosis (MS)[23], prostate cancer (PC)[24], rheumatoid arthritis (RA)[25], and type 2 diabetes (T2D)[26]. We computed the PGSs in three testing samples that were fully independent of the respective training samples (Table 1). For SCZ and MD, 62 and 22 testing cohorts, respectively, were used, and PGSs were computed based on the GWAS results that excluded the testing cohort from the Psychiatric Genomics Consortium (PGC). In evaluating the ICI, we concatenated all individual cohorts. Testing data from the UK Biobank[27] was used for BC, CAD, IBD, MS, PC, RA, and T2D. If SNP-wise $N_{\text{eff}}$ values were available in the GWAS results, the maximum $N_{\text{eff}}$ across all SNPs was used as input to the BPC approach (MD and SCZ). Alternatively, $N_{\text{eff}}$ was calculated as the sum of $N_{\text{eff}}$ of all contributing cohorts (CAD, IBD, MS, RA)[49]. If neither information was available, the SNP-wise $N_{\text{eff}}$ were estimated analytically with $N_{\text{eff}} = \frac{4}{2 \times AF \times (1 - AF) \times SE^2}$,[49] where AF = effect allele frequency and SE = standard error (PC, BC). Because the analytically derived $N_{\text{eff}}$ can produce large outliers, we used the 90th percentile across all SNPs instead of the maximum as input to the BPC approach.

Standard quality control was applied: Ambiguous (i.e., A/T or C/G SNPs), duplicate, and mismatching alleles for SNPs across training, testing, and population reference sample were removed[1]; a minor allele frequency filter of 10%, and, when available, an imputation INFO filter of 0.9 was applied as described before[31]; The major histocompatibility complex (MHC) was removed (hg19 coordinates: 6:28000000:34000000).

Posterior mean betas of SNPs were computed with PRScs-auto[12] (from here on simply referred to as PRScs; version June 4th, 2021) and SBayesR (version 2.03)[11]. PRScs uses a Linkage Disequilibrium (LD) reference panel based on HapMap3[57] SNPs and Europeans from the 1000 Genomes Project[28] (the default for PRScs). We use the default parameters listed on the software's GitHub page (https://github.com/getian107/PRScs). In the input of PRScs, we specified the sample size as $N_{\text{eff}}$ to ensure posterior mean betas were on the standardized observed scale with 50% case ascertainment. SBayesR uses an LD reference panel that is based on HapMap3[57] SNPs and 50,000 European UK Biobank subjects (the default for SBayesR version 2.03). In the input for SBayesR, we transformed the effect sizes to the standardized observed scale with 50% case ascertainment ($\beta_{50/50} = z/\sqrt{N_{\text{eff}}}$) and set the sample size to $N_{\text{eff}}$.

To estimate $R^2_{\text{liability}}$ we use an ancestry-matched population reference sample, namely the European sample of 1000 Genomes[28], which we downloaded from the MAGMA website (https://ctg.cncr.nl/software/magma).

The posterior mean betas were used to compute the PGS in 1000 Genomes and in the testing sample with Plink1.9 (version Linux 64-bit 6th June, 2021; command "--score <variant ID column > <effect allele column > <posterior mean beta> sum center"; https://doi.org/10.5281/zenodo.15721084).

The BPC approach requires a valid estimate of the prior disorder probability, which we set to the case-control ratio in the testing sample (see *Discussion* for approaches to estimate the prior disorder probability). We ascertained cases in the testing sample such that the case-control ratio was equal to 25%, 50%, or 75%.

### Reporting summary

Further information on research design is available in the Nature Portfolio Reporting Summary linked to this article.

## Data availability

Individual-level data from the Psychiatric Genomics Consortium (https://pgc.unc.edu/) and the UK Biobank (https://www.ukbiobank.ac.uk/enable-your-research/apply-for-access) cannot be shared freely, but an access application is required first. The GWAS summary statistics used in the UKB analyses can be requested or downloaded from the following web pages: Breast Cancer (https://bcac.ccge.medschl.cam.ac.uk/bcacdata/oncoarray/oncoarray-and-combined-summary-result/gwas-summary-associations-breast-cancer-risk-2020/); BMI (https://portals.broadinstitute.org/collaboration/giant/index.php/GIANT_consortium_data_files); Coronary Artery Disease (http://www.cardiogramplusc4d.org/data-downloads/#); Inflammatory Bowel Disease (https://www.ibdgenetics.org/); Multiple Sclerosis (https://imsgc.net/?page_id=31); Prostate Cancer (http://practical.icr.ac.uk/blog/?page_id=8164); Rheumatoid Arthritis (https://data.cyverse.org/dav-anon/iplant/home/kazuyoshiishigaki/ra_gwas/ra_gwas-10-28-2021.tar); Type 2 Diabetes (https://diagram-consortium.org/downloads.html). GWAS summary statistics for Major Depression and Schizophrenia can be downloaded from the PGC website (https://pgc.unc.edu/for-researchers/download-results/). 1000 Genomes reference files can be downloaded from https://ctg.cncr.nl/software/magma.

## Code availability

Scripts to apply the BPC approach can be downloaded from https://doi.org/10.5281/zenodo.15721084.

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

## Acknowledgements

We thank Naomi Wray, Peter Visscher, and Oliver Pain for their helpful discussions. D.P. is supported by the Netherlands Organization for Scientific Research—Gravitation project 'BRAINSCAPES: A Roadmap from Neurogenetics to Neurobiology' (024.004.012) and the European Research Council advanced grant 'From GWAS to Function' (ERC-2018-ADG 834057). A.L.P. has received an R01 grant from the US National Institutes of Health (HG006399). The PGC has received major funding from the US National Institute of Mental Health (PGC4: R01MH124839, PGC3: U01 MH109528; PGC2: U01 MH094421; PGC1: U01 MH085520). We thank the participants who donated their time, life experiences, and DNA to this research and the clinical and scientific teams that worked with them. We are deeply indebted to the investigators who comprise the PGC. Statistical analyses were carried out on the NL Genetic Cluster Computer (http://www.geneticcluster.org) hosted by SURFsara. The content is solely the responsibility of the authors and does not necessarily represent the official views of the National Institutes of Health.

## Author contributions

E.U.: Methodology, Software, Formal analysis, Investigation, Data Curation, Writing - Original Draft, Visualization. A.P.: Writing - Review & Editing D.P.: Writing - Review & Editing, Funding acquisition, Supervision. W.J.P.: Conceptualization, Methodology, Software, Resources, Writing – Original Draft and Review & Editing, Supervision.

## Competing interests

The authors declare no competing interests.

## Additional information

## Major Depressive Disorder Working Group of the Psychiatric Genomics Consortium

Cathryn M. Lewis [8,9] & Andrew M. McIntosh[8,9]

[8]Social, Genetic & Developmental Psychiatry Centre, Institute of Psychiatry, Psychology and Neuroscience, King's College London, London, UK. [9]Institute for Neuroscience and Cardiovascular Research, University of Edinburgh, Edinburgh, UK. A full list of members and their affiliations appears in the Supplementary Information.

## Schizophrenia Working Group of the Psychiatric Genomics Consortium

Micheal C. O'Donovan[10] & James T. R. Walters [10]

[10]Centre for Neuropsychiatric Genetics and Genomics, Cardiff University, Hadyn Ellis Building, Maindy Road, Cardiff, UK. A full list of members and their affiliations appears in the Supplementary Information.

