## [Transparent Peer Review file · Nature Communications]

Estimating Disorder Probability Based on Polygenic Prediction Using the BPC Approach

Corresponding Author: Mr Emil Uffelmann

Version 0:

Reviewer comments:

Reviewer #1

(Remarks to the Author)

In this work, Uffelmann et al. proposed a new approach to translate polygenic risk scores into absolute probabilities for disease outcomes. The authors showcased that this BPC approach outperforms one existing method in calibrating the absolute disease probability. This method is interesting, but I have several concerns.

1. Most importantly, the clinical utility of the BPC approach is likely limited. The authors stated the motivation of developing this method as: "With access to a sufficiently large population-representative tuning sample with relevant pheno- and genotype data, such a transformation can be achieved with existing methods[refs 5,6]. However, in most clinical settings, such samples are not readily available". Then, in empirical analysis, the authors created testing samples always with a case:control ratio of 1:1, and supplied the algorithm with a prior disorder probability of 50%. In practice, such a perfect specification is only possible when large-enough clinical samples are "readily available". The authors should show the impact of mis-specification of the prior disorder probability in empirical analysis.

2. Related to #1, to date, very few polygenic risk scores have demonstrated outstanding predictive performance in distinguishing cases from controls, which is the very reason why the proposed use of such scores has mainly focused on risk stratification in large populations instead of generating individualized risks. This means that, in Equation (3), $P(\text{PGS}_i | D_i = \text{case})$ can be very close to $P(\text{PGS}_i)$ for many scores, thus the output of the BPC approach can be largely driven by the prior. The authors should investigate the distributions of $P(\text{PGS}_i | D_i = \text{case}) / P(\text{PGS}_i)$ in both simulations and empirical analysis to show to what extent the results are driven by the prior.

3. The authors should apply one or more individual-level data-based calibration methods, such as those in refs 5 and 6, as positive controls. While the BPC approach may not outperform these methods, such results can help to benchmark the ICI and other metrics.

4. The method assumes that the polygenic risk score are well-calibrated on the liability scale. How reasonable is this assumption?

5. The prior disorder probability can be individualized, but the authors have not explored or discussed this potential. Can family history (PMID: 35935918; 35710731) or other risk factor profiles help to better predict the disease risk?

6. The method has not considered the age of disease onset, which is more challenging statistically but probably much more clinically impactful.

7. Many more mathematical details are needed in the Methods section for the audience to better understand the underlying rationale, in addition to citations of existing studies. These could be in Supplementary Notes:

(1) Step 1: Justification is needed for "The BPC approach requires the posterior mean betas to be on the standardized observed scale with 50% case ascertainment"; Why does $z/\sqrt{N_{\text{eff}}}$ give the desired posterior means? Why does $N_{\text{eff}} = 4 / (1 / N_{\text{case}} + 1 / N_{\text{control}})$ or the empirical estimate $N_{\text{eff}} = 4 / (2 * AF * (1 - AF) * SE^2)$ work? What is the connection between these estimates and those obtained for continuous traits?

(2) Step 2: Equation (1) should be elaborated - how does the formula convert the posterior means to the liability scale? How is the factor of $K * (1 - K) / (z * 0.5)$ derived?

(3) Step 3: Details are needed for "Given $R^2_{liability}$, the expected mean and variance of the PGS can be estimated in cases and in controls using normal theory"; How is this done specifically?

(Remarks on code availability)

The code is a usable resource for the community with relatively detailed instructions.

Reviewer #2

(Remarks to the Author)

This manuscript presents a method BPC to compute the absolute disorder probability from polygenic score (PGS). The work is important to the field and potential clinical implementation of PGS. They show that BPC outperforms another method when computing the probability. The analysis overall looks good to me and I only have a few comments regarding the assumption of the methodology.

1. In step 2, it is assumed that the case ascertainment ratio is 50%. Is this based on the ratio of cases in GWAS summary statistics? If so, do you need to change the ratio based on the case control ratio in the GWAS discovery cohort?

2. The simulation setting is unrealistic as it only includes 1000 SNPs in linkage equilibrium. In practice, a much larger number of SNPs with LD is expected to be included in a PGS model.

3. Assumption of normality. The derivation in step 4 makes use of the normality assumption (or truncated normal) of PGS to calculate $P(\text{PGS}_i | D_i = \text{case})$. Based on my experience, the distribution of PGS sometimes has a heavier tail than normal distribution. Would this affect your calculation of disease probability?

4. Calibration of PRS. It is assumed that PGS is well-calibrated with slope being 1. Have the authors checked from the real data that this assumption is satisfied? At least from the simulation studies of PRS-CS paper, the slope can sometimes have large deviation from 1 (Supplementary Table 7 from https://static-content.springer.com/esm/art%3A10.1038/s41467-019-09718-5/MediaObjects/41467_2019_9718_MOESM1_ESM.pdf).

5. In line 16 and 20 from BPC code (<https://github.com/euffelmann/bpc/blob/main/bpc.R>), the variance of PGS is calculated as $r^2 - k_1 r^2 * r^2$. Should it be $(1 - k_1) r^2$?

6. $R^2_{liability}$ is derived from a population reference panel and crucial for the model.

Does it make an implicit assumption that the number of SNPs is the same among the reference panel used to calculate $R^2_{liability}$, testing cohort and PRS model. Otherwise the scale of PGS may be different when applying to different cohorts and may affect the calculation of probability.

(Remarks on code availability)

Version 1:

Reviewer comments:

Reviewer #1

(Remarks to the Author)

The authors have addressed my comments.

(Remarks on code availability)

Reviewer #2

(Remarks to the Author)

The authors have addressed my previous concerns.

(Remarks on code availability)

Response to reviewers for NCOMMS-24-02174 (Uffelmann et al.)

REVIEWER COMMENTS

Reviewer #1 (numbers added to reviewer comments: R1C1, R1C2, etc.)

(Remarks to the Author):

In this work, Uffelmann et al. proposed a new approach to translate polygenic risk scores into absolute probabilities for disease outcomes. The authors showcased that this BPC approach outperforms one existing method in calibrating the absolute disease probability. This method is interesting, but I have several concerns.

R1C1. Most importantly, the clinical utility of the BPC approach is likely limited. The authors stated the motivation of developing this method as: "With access to a sufficiently large population-representative tuning sample with relevant pheno- and genotype data, such a transformation can be achieved with existing methods[refs 5,6]. However, in most clinical settings, such samples are not readily available". Then, in empirical analysis, the authors created testing samples always with a case:control ratio of 1:1, and supplied the algorithm with a prior disorder probability of 50%. In practice, such a perfect specification is only possible when large-enough clinical samples are "readily available". The authors should show the impact of mis-specification of the prior disorder probability in empirical analysis.

The reviewer stated that the clinical utility of the BPC approach is likely limited based on three related comments: (i) the authors created testing samples always with a case:control ratio of 1:1 (ii) perfect specification of the prior is only possible when large enough samples are available, and (iii) empirical analysis based on misspecification of the prior should be shown. We address these three comments in turn.

(i) The authors created testing samples always with a case:control ratio of 1:1
We agree that our results mainly present results for correctly specified prior of 1:1 (i.e., 0.50). We also show that BPC is well-calibrated for a range of correctly specified priors (0.25, 0.5, and 0.75; see Figure 3), which we now describe more clearly in subsection "Empirical Analyses" of the Results (p. 18) and Discussion (p. 25).

(ii) Perfect specification of the prior is only possible when large enough samples are available
We agree that estimating a prior in practice can be challenging and that we did not describe this clearly in our previous manuscript. While challenging, there are several ways to obtain a prior. First, a small clinical reference sample may be used that does not require genotype data and can be smaller than the samples required for the tuning approaches (i.e., samples that are much easier to obtain than those needed for tuning approaches of refs (Ashenhurst et al., 2021; Sun et al., 2021); also see R1C3). Second, context-specific prevalence estimates may be obtained from the literature (see ref (Zaitlen et al., 2012)). Third, prior elicitation may be applied, whereby a single clinician (or a panel of clinicians) provides a subjective estimate of the prior. Fourth, if considerable uncertainty about the prior exists, one may use a range of priors in the BPC approach to obtain a range of posterior disorder probabilities. We have updated the Abstract, subsection "Input" of the Methods (p. 5), Discussion (p. 25), and the GitHub manual (<https://github.com/euffelmann/bpc>).

(iii) Empirical analysis based on misspecification of the prior should be shown

We have now added a new supplementary figure with nine panels for each analyzed phenotype (Supplementary Figure 18), showing the effects of misspecifying the prior in empirical analyses. For each of the nine phenotypes, we varied the true and assumed prior between 0.25, 0.50, and 0.75, plotting the results for all nine permutations. These analyses demonstrate that the mean predicted disorder probabilities closely follow the assumed prior and that misspecifying the prior (i.e., the true and assumed prior are discordant) negatively affects the Integrated Calibration Index (ICI). We now quantify the impact of misspecifying the prior on the ICI in the Results. However, we also show that the BPC approach is well calibrated at different correctly specified values of the prior (i.e., 0.25, 0.50, and 0.75; see Main Figure 3 and Supplementary Figure 18), which suggests that the change of the posterior disorder probability to the prior remains informative across a range of priors. We note that we used a prior of 0.5 throughout the paper for convenience. We have updated subsection “*Input*” in the Methods (p. 5), subsection “*Empirical analyses*” in the Results (p.21), and the Discussion (p.26).

In summary, the BPC approach is well-calibrated across a range of correctly specified priors, and, while challenging, the prior can be specified in a number of empirical ways, and uncertainty can be incorporated by specifying a range of priors. Thus, we believe the BPC approach has appreciable potential to have clinical utility (also see R1C2).

R1C2. Related to #1, to date, very few polygenic risk scores have demonstrated outstanding predictive performance in distinguishing cases from controls, which is the very reason why the proposed use of such scores has mainly focused on risk stratification in large populations instead of generating individualized risks. This means that, in Equation (3), $P(\text{PGS}_i | D_i = \text{case})$ can be very close to $P(\text{PGS}_i)$ for many scores, thus the output of the BPC approach can be largely driven by the prior. The authors should investigate the distributions of $P(\text{PGS}_i | D_i = \text{case}) / P(\text{PGS}_i)$ in both simulations and empirical analysis to show to what extent the results are driven by the prior.

The reviewer has made two comments: (i) very few polygenic scores have outstanding predictive performance, and (ii) distributions of $P(\text{PGS}_i | D_i = \text{case}) / P(\text{PGS}_i)$ should be investigated. We address these comments in turn.

(i) Very few polygenic scores have outstanding predictive performance

While the reviewer is correct that most PGSs have low predictive performance, there are several phenotypes for which PGSs exhibit predictive performance that rivals that of other clinically useful predictors. These include coronary artery disease (Khera et al., 2018), breast cancer (Mavaddat et al., 2019), chronic obstructive pulmonary disease (Zhang et al., 2025) and prostate cancer (Mars et al., 2020). With ever-increasing GWAS sample sizes, we expect the PGSs for other phenotypes will improve in predictive performance over the next years. We have updated the description of these considerations in the Introduction (p. 3) and Discussion (p.26).

(ii) Distributions of $P(\text{PGS}_i | D_i = \text{case}) / P(\text{PGS}_i)$ should be investigated

We thank the reviewer for their suggestion. We have conducted the analyses as suggested and now include Supplementary Figure 14 (pertaining to simulation) and Supplementary Figure 24 (pertaining to empirical analyses), and we have updated the subsections ‘Simulation analysis’ (p.18) and ‘Empirical analysis’ (p.22) of the Results. In short, we found that the distribution varies markedly around 1 for most realistic conditions in simulation analyses (e.g., S.D. = 0.3 for $K = 0.01$, $R^2_{\text{liability}} = 0.05$, and prior = 0.50) and for all traits we studied in our empirical analyses (e.g., S.D. = 0.29 for schizophrenia when the prior = 0.50), demonstrating that the posterior predicted disorder probabilities differ markedly from the prior.

R1C3. The authors should apply one or more individual-level data-based calibration methods, such as those in refs 5 and 6, as positive controls. While the BPC approach may not outperform these methods, such results can help to benchmark the ICI and other metrics.

We thank the reviewer for their suggestion. We have added two methods using tuning samples.

The first method we consider (*Logit-tuned*) computed predicted disorder probabilities with a method outlined in a 23andme whitepaper (Ashenhurst et al., 2021) based on logistic regression of PGS on disorder status in a tuning set independent from both the training GWAS and the testing set. The second method we consider (*BPC-tuned*) applies a modified version of the BPC approach that empirically estimates the distribution of PGSs in cases and controls in a tuning sample instead of estimating it theoretically from $R^2_{liability}$.

We find little difference between the BPC approach and those using tuning samples at tuning samples up to 200 cases and 200 controls. At larger tuning sample sizes, the tuning methods have smaller ICI than the BPC approach (i.e., approximately 0.015 smaller). However, we consider BPC's calibration (ICI < 0.03) satisfactory, such that the BPC approach will likely be preferred in most settings as it does not require collecting large tuning samples.

We have added the new main Figure 5, added a new subsection, '*Approaches using tuning samples*' to the Method section (p. 9), a new subsection, '*Calibration of tuning approaches*' to the Results section (p. 22), and updated the Discussion (p. 25).

R1C4. The method assumes that the polygenic risk score are well-calibrated on the liability scale. How reasonable is this assumption?

Our response is threefold.

First, in simulation, we explicitly simulated the liability for each individual, meaning we can assess the validity of our assumption that the PGS (based on Bpred, a version of LDpred without LD (Vilhjálmsón et al., 2015)) is well-calibrated on the liability scale. We validate that the slope and intercept are, on average, 1 and 0, respectively, when regressing the liability on the PGS. We have added Supplementary Figure 13 and updated the Results section (p. 18).

Second, for empirical analyses, we note that "liability" refers to an unobserved continuous risk underlying binary disease phenotypes. Consequently, one cannot directly test if the PGS is on the liability scale for the diseases considered. However, given that the BPC approach yields well-calibrated results on the observed scale, this indirectly indicates that the PGS is well-calibrated on the liability scale. We have updated the Results section (p.19) to include this observation.

Third, when the PGS is well-calibrated on the liability scale, it follows that the variance of the PGS in a population-based sample (such as 1000G) is equal to the explained variance of the PGS in the phenotype (see equation 2 in the manuscript). In Figure 6, we show that this holds in simulation and empirical analyses, providing indirect support that PGSs are well-calibrated on the liability scale. We have updated the "*Estimation of variance explained ($R^2_{liability}$)*" subsection of the Results section (p.23) to include this observation.

In conclusion, we believe that the assumption that the PGSs are well-calibrated on the liability scale is valid, as evidenced in simulation and empirical analyses.

We provide a similar reply to reviewer 2 (see R2C4).

R1C5. The prior disorder probability can be individualized, but the authors have not explored or discussed this potential. Can family history (PMID: 35935918; 35710731) or other risk factor profiles help to better predict the disease risk?

We agree that we did not discuss the potential of individualizing the prior in detail. We have updated the Discussion section (p. 27).

Specifically, we distinguish between updating the prior based on factors correlated to the PGS and factors not correlated to the PGS.

For factors not correlated to the PGS, such as age (also see R1C6) and sex, the BPC approach can be updated by adjusting the prior. We updated the Discussion section (p. 27).

For factors correlated to the PGS, such as family history (PMID 35935918 and 35710731: Hujoel et al., 2022; Lu et al., 2022) (Krebs, Appadurai, et al., 2023; Krebs, Hellberg, et al., 2023), the BPC approach does not currently include a direct way to include this in the framework. This is because, in this case, the prior cannot be adjusted independently without modifying the $R^2_{liability}$. We agree that this is an important direction for future research, and we added it to the Discussion section (p.27).

R1C6. The method has not considered the age of disease onset, which is more challenging statistically but probably much more clinically impactful.

As suggested above, a participant's age can be incorporated by adjusting the prior. We agree that incorporating age directly into the BPC approach is an important direction for future research, as we now highlight in the Discussion (p. 27).

Ideally, the probabilities are reported as a function of the individual's age at assessment and the distribution of the age-of-onset of the disorder to also model the decline in risk in the years following assessment in which the disorder has not manifested itself. This requires more advanced statistical extensions of the BPC approach and detailed prevalence and incidence rates for all age groups. We agree that this would further enhance the clinical utility of the BPC approach. We have updated the Discussion section (p.27) to clarify the interpretation of probabilities computed with the current BPC approach and to emphasize this important direction for future research.

R1C7. Many more mathematical details are needed in the Methods section for the audience to better understand the underlying rationale, in addition to citations of existing studies. These could be in Supplementary Notes:

(1) Step 1: Justification is needed for "The BPC approach requires the posterior mean betas to be on the standardized observed scale with 50% case ascertainment"; Why does $z/\sqrt{N_{eff}}$ give the desired posterior means? Why does $N_{eff} = 4 / (1 / N_{case} + 1 / N_{control})$ or the empirical estimate $N_{eff} = 4 / (2 * AF * (1 - AF) * SE^2)$ work? What is the connection between these estimates and those obtained for continuous traits?

We thank the reviewer for pointing out this lack of detail in our manuscript. We have added Supplementary Notes 1 and 2 and refer to this in the subsection "input" (p. 5) and "Step 1

Compute posterior mean betas with a Bayesian PGS method” of the Methods (p. 6), where we provide an answer to each of these questions.

(2) Step 2: Equation (1) should be elaborated - how does the formula convert the posterior means to the liability scale? How is the factor of $K * (1 - K) / (z * 0.5)$ derived?

We have added Supplementary Note 3 and refer to this in subsection “Step 2 Transform posterior mean betas to liability scale” of the Methods (p. 6).

(3) Step 3: Details are needed for "Given $R^2_{liability}$, the expected mean and variance of the PGS can be estimated in cases and in controls using normal theory"; How is this done specifically?

We have added Supplementary Note 4 and refer to this in the subsection “Step 3 Derive $R^2_{liability}$ and the expected distribution of the PGS in cases and controls” of the Methods (p. 7).

Reviewer #1 (Remarks on code availability):

The code is a usable resource for the community with relatively detailed instructions.

Reviewer #2 (numbers added to reviewer comments: R2C1, R2C2, etc.)
(Remarks to the Author):

This manuscript presents a method BPC to compute the absolute disorder probability from polygenic score (PGS). The work is important to the field and potential clinical implementation of PGS. They show that BPC outperform another method when computing the probability. The analysis overall looks good to me and I only have a few comments regarding the assumption of the methodology.

We thank the reviewer for suggesting that our work is important to the field and for the potential clinical implementation of PGS. We respond to the reviewer’s comments in turn.

R2C1. In step 2, it is assumed that the case ascertainment ratio is 50%. Is this based on the ratio of case in GWAS summary statistics? If so, do you need to change the ratio based on the case control ratio in the GWAS discovery cohort?

We thank the reviewer for their question. We have updated Figure 1 and the subsection “Step 1” of the Methods (p.6).

Specifically, the case ascertainment ratio of 50% does not need to be adjusted to the case-control ratio in the discovery GWAS for step 2. Rather, in step 1 of the BPC approach, posterior mean betas are computed on the standardized observed scale with 50% case ascertainment based on a statistical transformation. PRSCs achieves this by using the effective sample size (N_{eff}) as input (as opposed to the total sample size; also see R1C7 (1) and Supplementary Note 1). For SBayesR, an additional transformation is required) to first transform betas to the standardized observed scale $\beta_{50/50} = z / \sqrt{N_{eff}}$ (see Methods Step 1; see R1C7 and Supplementary Note 2). Because of this transformation in Step 1, the case ratio in Step 2 is correctly scaled at 50%. We have updated Figure 1, the subsection “Step 1” of the Methods (p. 6), and we have added Supplementary Notes 1 and 2 to provide more details (also see R1C7).

We also note that the prior disorder probability used in computing the posterior disorder probability (Step 4 of Figure 1) is independent of the discovery GWAS results, and can be specified flexibly. We have clarified this further in the subsection “Step 4” of the Methods (p.7).

R2C2. The simulation setting is unrealistic as it only includes 1000 SNPs in linkage equilibrium. In practice, much larger number of SNPs with LD is expected to be included in a PGS model.

The reviewer correctly notes that we used a simplified simulation setup. We recognize we did not clearly justify this choice.

We note that we simulated a small number of SNPs (M) as this allows simulating smaller training sample sizes (N), thereby limiting computational cost because the PGS R^2 primarily depends on M/N , such that simulations at reduced values of both M and N are appropriate (Vilhjálmsón et al., 2015). We describe this in the subsection “Simulation analysis” in the Results (p. 14).

Additionally, in the revised manuscript, we now include additional simulation analyses doubling both M and N (resulting in the same variance explained by the PGS, which depends on M/N). We found that this did not change our findings. We have updated the subsection “Simulation analysis” of the Results (p. 16).

We simulated no LD to limit computational costs because LD has no impact on the scale of the polygenic score, which aggregates all SNP effects together into a single score. We have updated the subsection “Simulation analysis” of the Methods (p. 11) and Results section (p. 14) to clarify our choice. If the reviewer and the editor see significant added value in including simulation also including LD, we will be happy to add these.

We further note that our simulation and empirical analyses lead to the same conclusions, and we thus believe that our current simulations are appropriate to validate the BPC approach. We now mention the concordance of simulation and empirical results explicitly in the Discussion (p. 25).

R2C3. Assumption of normality. The derivation in step 4 makes use of the normality assumption (or truncated normal) of PGS to calculate $P(\text{PGS}_i | D_i = \text{case})$. Based on my experience, the distribution of PGS sometimes has a heavier tail than normal distribution. Would this affect your calculation of disease probability?

We thank the reviewer for suggesting this sensitivity analysis to probe our assumptions. Our response is threefold.

First, we have empirically tested whether the PGS distribution significantly deviates from the expected normal distribution using the Kolmogorov-Smirnov test, which quantifies the distance between the normal distribution and the observed distribution (Massey, 1951). The Kolmogorov-Smirnov test for normality is not significant for any of the PGSs of the studied phenotypes (see Supplementary Figure 23), which we note in the ‘*Empirical analysis*’ subsection of the results (p.22) and in the Discussion section (p. 27).

Second, we agree with the reviewer that analytically, the PGSs in cases and controls are expected to be not 100% perfectly normally distributed due to the ascertainment and truncating of the normally distributed liability. We now include simulations to explore this phenomenon, and we observed that deviating from normality only starts for PGS explaining >60% of variance on the liability scale (see Supplementary Figures 12), which we note in the ‘*Simulation analysis*’ subsection of the Results (p. 17). We note that in our empirical analyses, $R^2_{\text{liability}}$ ranges from 3-15% (see Figure 6).

Third, we note that for disorders with a couple of common, very large-effect causal variants (e.g., the APOE for Alzheimer’s Disease), the normality assumption may be violated. We discuss this in the limitation paragraph of our revised manuscript and highlight integrating prediction based on rare variants with large effects as an important direction for future research (p. 27).

In conclusion, given the parameters and traits considered, the assumption of normality is valid in both our simulation and empirical analyses.

R2C4. Calibration of PRS. It is assumed that PGS is well-calibrated with slope being 1. Have the author checked from the real data that this assumption is satisfied? At least from the simulation studies of PRS-CS paper, the slope can sometimes have large deviation from 1 (Supplementary Table 7 from https://static-content.springer.com/esm/art%3A10.1038/s41467-019-09718-5/MediaObjects/41467_2019_9718_MOESM1_ESM.pdf).

The reviewer makes two related comments: (i) Have the authors checked if the PGSs are well-calibrated with a slope of 1, and (ii) the slopes in the PRS-CS paper can have large deviations from one. We respond to these comments in turn.

(i) Have the authors checked if the PGS are well-calibrated with a slope of 1
We assume the reviewer is referring to calibration on the continuous liability scale, and our response is threefold.

First, in simulation, we explicitly simulated the liability for each individual, meaning we can assess the validity of our assumption that the PGS is well-calibrated on the liability scale (on average across 100 simulation runs). We validate that the slope and intercept are on average 1 and 0, respectively. We have added Supplementary Figure 13 and updated the subsection “Simulation analysis” in the Results (p. 18).

Second, for empirical analyses, we note that “liability” refers to an unobserved continuous risk underlying binary disease phenotypes. Consequently, one cannot directly test if the PGS is on the liability scale. However, given that the BPC approach yields well-calibrated results on the observed scale, this indirectly suggests that the PGS is well-calibrated on the liability scale. We have updated the “Empirical analysis” subsection of the Results section (p.19) to include this observation.

Third, when the PGS is well-calibrated on the liability scale, it follows that the variance of the PGS is equal to the explained variance of the PGS on the liability scale (see equation 2 in the manuscript). In Figure 5, we show that this holds in simulation and empirical analyses, providing indirect support that PGSs are well-calibrated on the liability scale. We have updated the “*Estimation of variance explained ($R^2_{liability}$)*” subsection of the Results section (p.23) to include this observation.

We provide a similar reply to reviewer 1 (see R1C4).

(ii) The slopes in PRS-CS paper can have large deviations from one.

PRS-CS-auto slopes can indeed have large deviations from 1 (see Supplementary Table 7 in (Ge et al., 2019)). However, it’s important to note that the values in the PRS-CS paper are reported for single simulation runs, while we report the mean of the slopes across 100 simulation runs.

The values reported in the PRS-CS paper fit within the distributions we observe (see Supplementary Figure 13).

It should be noted that the ICI has been deemed a better metric of calibration for disorder traits because the ICI is robust to sparse subregions of poor calibration (Austin et al., 2019), which is why we report the ICI as the primary metric of calibration. We have updated the subsection “Metrics of performance” in the Methods to describe this more clearly (p.10).

In additional simulation analyses, we compared the properties of the ICI and the calibration slope. We indeed observed that the ICI is more stable than the calibration slope, specifically for PGS explaining < 10% of variance (Supplementary Figure 7). We have included Supplementary Figure 7 and updated the Results section (p. 16) of the revised manuscript.

R2C5. In the line 16 and 20 from BPC code (<https://github.com/euffelmann/bpc/blob/main/bpc.R>), the variance of PGS is calculated as $r^2 - k_1 r^2 * r^2$. Should it be $(1 - k_1) r^2$?

We thank the reviewer for looking closely at our code. The original code is correct. We anticipate the confusion may have come from some redundant code in the function (e.g., $k_1 <- i_1 * (i_1 - t); 1 - k_1$). The (“1- k1”) was a print statement that should have been deleted. We have cleaned up the code and deleted the redundant statements after the semicolons. The formulas follow from refs (Bulmer, 1980; Tallis, 1987).

We also note that we now provide a detailed Supplementary Note 3 describing every step of the method (also see R1C7).

R2C6. R2_liability is derived from a population reference panel and crucial for the model. Does it make implicit assumption that the number of SNPs is same among the reference panel used to calculate R2_liability, testing cohort and PRS model. Otherwise the scale of PGS may be different when applying to different cohorts and may affect the calculation of probability.

It is indeed important that the same SNPs are used in the testing cohort, the PGS model (i.e., the training sample), and the population reference sample. If the SNPs in the PGS model do not precisely match the SNPs in the testing cohort, prediction can be negatively affected. This is because Bayesian PGS methods optimize effect sizes based on LD between specific sets of SNPs. Their omission can strongly impact prediction. We have updated the subsection “Input” of the Method section (p. 5) and the BPC manual (<https://github.com/euffelmann/bpc>) to be more explicit about this.